# A Microfluidic Platform to Monitor Real-Time Effects of Extracellular Vesicle Exchange between Co-Cultured Cells across Selectively Permeable Barriers

**DOI:** 10.3390/ijms23073534

**Published:** 2022-03-24

**Authors:** Hunter G. Mason, Joshua Bush, Nitin Agrawal, Ramin M. Hakami, Remi Veneziano

**Affiliations:** 1School of System Biology, George Mason University, Manassas, VA 20110, USA; hmason4@gmu.edu; 2Center for Infectious Disease Research, George Mason University, Manassas, VA 20110, USA; 3Department of Bioengineering, College of Engineering and Computing, George Mason University, Manassas, VA 20110, USA; jbush20@gmu.edu (J.B.); nagrawal2@cnmc.org (N.A.)

**Keywords:** microfluidic device, lab-on-a-chip, intercellular communication, exosomes, extracellular vesicles, functional EV assays, extracellular matrix, Matrigel, PEGDA

## Abstract

Exosomes and other extracellular vesicles (EVs) play a significant yet poorly understood role in cell–cell communication during homeostasis and various pathological conditions. Conventional in vitro and in vivo approaches for studying exosome/EV function depend on time-consuming and expensive vesicle purification methods to obtain sufficient vesicle populations. Moreover, the existence of various EV subtypes with distinct functional characteristics and submicron size makes their analysis challenging. To help address these challenges, we present here a unique chip-based approach for real-time monitoring of cellular EV exchange between physically separated cell populations. The extracellular matrix (ECM)-mimicking Matrigel is used to physically separate cell populations confined within microchannels, and mimics tissue environments to enable direct study of exosome/EV function. The submicron effective pore size of the Matrigel allows for the selective diffusion of only exosomes and other smaller EVs, in addition to soluble factors, between co-cultured cell populations. Furthermore, the use of PEGDA hydrogel with a very small pore size of 1.2 nm in lieu of Matrigel allows us to block EV migration and, therefore, differentiate EV effects from effects that may be mediated by soluble factors. This versatile platform bridges purely in vitro and in vivo assays by enabling studies of EV-mediated cellular crosstalk under physiologically relevant conditions, enabling future exosome/EV investigations across multiple disciplines through real-time monitoring of vesicle exchange.

## 1. Introduction

In addition to normal physiological crosstalk, the progression of a variety of disease states is largely dependent on the secretion and subsequent uptake of bioactive extracellular vesicles (EVs) [1]. These vesicles are heterogeneous in nature, and can range from 30 nm up to 10 μm in size [2]. Among these EVs, exosomes are a key player in the EV-based crosstalk that regulates disease progression [3,4]. Exosomes are a subtype of EVs that typically range from 30 to 150 nm in diameter, and are secreted by nearly every cell type in the human body [5,6]. Their membrane and lumen are enriched with bioactive cargo molecules such as proteins, lipids, and nucleic acids that can be delivered to neighboring or distant recipient cells [5,7,8]. Over the past decade, exosomes have been shown to play a major modulatory role that can either promote or inhibit the progression of many human diseases, such as cancer, neurological disorders, cardiovascular diseases, and infectious diseases [3,9,10,11,12,13,14,15]. The potential of exosomes for use as diagnostic biomarkers or for the design of effective therapeutics and vaccines is being intensely investigated, given their potent regulatory function, availability in most bodily fluids, and incorporation of various disease-specific cargoes [16]. Despite this significance and potential, progress in the mechanistic understanding of exosome functions and relevance across many specific diseases is still extremely limited, and much remains to be explored. In part, this is due to the difficulty of working with exosomes (and other EVs), both in vitro and in vivo. For instance, the gold standard for performing in vitro exosome studies requires a lengthy and tedious series of centrifugation steps that includes density gradient separations to purify exosomes from the culture supernatant. Furthermore, large amounts of producer cells and culture media are needed per isolation to recover sufficient material [17]. Alternatives that also offer reliable isolation—such as size-exclusion chromatography—are available, but are relatively costly and typically not able to sufficiently enrich the vesicles to obtain suitable yields for some applications, or to provide sufficient separation of exosomes from larger EVs [18].

Likewise, in vivo exosome studies are inefficient. Due to their size and the complex tissue environment, direct observation of exosomes in vivo is not easily achievable [19]. Furthermore, it is not yet readily possible to effectively target functionalized exosomes to distinct cell types in vivo in order to monitor specific interactions and functions without potential complications arising from exosome alteration [20]. In addition to circumventing these limitations, it is highly desirable to have physiologically relevant functional assays that bridge purely in vitro and in vivo experimental designs and more closely replicate the in vivo setting, where exosomes/EVs are constitutively exchanged between donor and recipient cells [21]. Microfluidic technology has been extensively utilized to attempt to alleviate some of the challenges associated with EV studies. This includes enhanced sorting, detection, and isolation of EVs [18], efficient surface modification [22], and use in extrusion for the generation of new vesicles [23]. However, current microfluidic technology is limited with regard to simulating native vesicle exchange between co-cultured cells in order to enable functional studies.

To address this need, we developed a microfluidic lab-on-a-chip-based approach to enhance functional EV studies. Our work adapted chemotactic cell migration chips that are based on the presence of several individual cell chambers separated by thinner ribbed channels [24,25,26]. We filled the rib channels with diffusion barrier hydrogels mimicking the extracellular matrix (ECM), such as Matrigel or poly(ethylene glycol) diacrylate (PEGDA). Matrigel has been previously used to embed cells in order to enable EV secretion through their permeable matrix for functional analysis or collection [21,27]. Instead, in our design, Matrigel was employed as a selectively permeable barrier that permits diffusion of small EVs and soluble factors between separated cell populations, allowing our system to serve as a functional EV platform. This system simulates the ECM–cell interplay of an in vivo environment that allows for functional studies under physiologically relevant EV production and distribution conditions, while removing the need for the often tedious and lengthy EV purification procedures. The small scale of the chip and its micro-sized channels helps to avoid the resource-intensive setups typically needed, requiring very few cells and only a few microliters of media volume, and permits the chips to be bonded to standard-size coverslips or glass slides for simple microscope-based observation of the vesicles and cell populations. Our system fully integrates cell culturing with microfluidic EV manipulation to create a novel in vitro EV assay. This system will provide a versatile tool for investigators across multiple disciplines to perform functional EV studies under physiologically relevant conditions.

## 2. Results

### 2.1. Microfluidic Design

A five-channel microfluidic system was built as described by Roberts et al. [25], with three cell culture lanes separated by two matrix channels, which we utilize as selective diffusion barriers between each cell culture lane (Figure 1a,b). The matrix channels of the assembled microchip were ribbed to enable facile injection of hydrogel by modulating the surface wettability of the matrix channel. The versatility of the design and protocol allows injection of a variety of hydrogels with different pore sizes that prevent passive cell migration and grant control of the size-based particle diffusion selection (Figure 1c). This can be utilized to study different-sized EV populations, or to block EV migration altogether, so as to distinguish functional effects of EVs from soluble-factor-mediated effects. Additionally, a truncated two-cell lane alternative was generated containing only two cell channels and a single matrix channel, and was utilized for the studies reported here, as it did not require the accessory channel (Figure 1a).

### 2.2. Production and Optimization of Microfluidic Chips

The steps for manufacturing the microfluidic chips are presented in Figure 2, and further described in the Materials and Methods section. The following steps (Figure 2a) improved the gel loading efficiency within the matrix channels: (1) optimization of PDMS layer thickness and creation of the injection ports; (2) use of a handheld corona treater (Figure 2b); and (3) precise activation of a single channel using a directed electrical pulse. Specifically, we utilized a corona treater in place of flood oxygen–plasma exposure, thus limiting the required equipment and expediting the assembly process. The selective loading of matrix channel was achieved by balancing the hydrodynamic pressure and surface tension of the gel. The interconnecting ribs were designed significantly narrower than the main gel channel in order to allow the gel infusion to be guided by capillary action, while the surface tension at the open end would prevent it from leaking into the cell chambers. To further improve Matrigel–PDMS interaction, the matrix channels were plasma activated to enhance their hydrophilicity and bonding to the glass substrate. First, the microchannels were rendered hydrophobic by heating the devices at 200 °C for 1 h (Figure 2c). Subsequently, a handheld corona treater (Figure 2c) was utilized to selectively activate the matrix channel for efficient Matrigel loading. The gel infusion strategy was further augmented by temporarily placing a ground electrode at the opposite end of this channel (Figure 2c), while blocking the other inlets/outlets, in order to allow an electric pulse of greater power to be used without it spreading out from the matrix channel. The spread of the electric pulse beyond the channel ribs would activate the adjacent channels, thus causing the Matrigel injection to leak beyond its respective channel. This approach utilizes back pressure, generated by local heating of the plasma pulse concentrated in the Matrigel ribs, as a barrier that allows the plasma to be directed through the matrix channel specifically.

### 2.3. Matrigel Diffusion Characterization

To determine the effective diffusible particle size profile of Matrigel at our working concentration of 8 mg/mL, we imaged diffusion of different-sized fluorescent particles across the gel matrix. The effective pore size for Matrigel has been previously shown to be ~140 nm, ranging to upwards of 350 nm [28]. We used liposomes to mimic diffusion of EVs across the Matrigel, which can be prepared to have specific diameter ranges. Liposomes fluorescently labeled with Cy5 were prepared by extrusion to produce monodisperse population sizes of 70 nm and 250 nm (Appendix A). The liposome solutions were manually injected into the donor channels of Matrigel-loaded chips (Figure 3a,b). Diffusion of the vesicles was imaged and compared between post-injection time points of 20 min and 24 h. Both vesicle populations were able to diffuse through within the 24 h period. As expected due to their smaller size, the 70 nm vesicles diffused through more rapidly than the 250 nm ones when compared at the 24 h time points (Figure 3a,b, recipient channel). To find an upper limit of the diffusion profile, 500 nm yellow–green fluorescing polystyrene beads were also injected into the donor channel of a Matrigel-loaded chip, and the diffusion was imaged again at 20 min and 24 h time points. There was no observed diffusion of the beads even at the 24 h time point (Figure 3c). Therefore, based on our results, vesicles with diameters of at least 250 nm, but less than 500 nm, can pass through Matrigel. This should allow diffusion of exosomes and other smaller EVs but prevent the crossing of larger biological particles, such as apoptotic bodies, larger microvesicles, and oncosomes, or the passage of bacteria and cells [29,30,31].

### 2.4. Exosome Diffusion in the Microfluidic Chip

We next verified that exosomes can diffuse from the donor channel to the recipient channel and be subsequently taken in by the recipient cells. The Hakami laboratory has used monocytic U937 cells during extensive mechanistic studies of how exosomes regulate innate immune responses to infection, e.g., [4]. Therefore, we used this cell line for our chip characterization studies. Exosomes were recovered from U937 cells and, after labelling with Cy5 fluorescent lipids, were purified using the density gradient procedure detailed in the Materials and Methods section. Purified, labeled exosomes were injected into the donor channel of a Matrigel-loaded chip, and the chips were imaged at 20 min and 24 h time points after injection (Figure 4a). Successful diffusion of exosomes was observed across the gel channel. This was also repeated with a PEGDA-hydrogel-filled chip with an expected pore size of 1.2 nm [32], showing—as was expected—that there was no diffusion of exosomes into the hydrogel after 24 h (Appendix A). We then injected U937 cells into the recipient channel of a new Matrigel-loaded chip using a syringe pump, and injected Cy5-labeled exosomes into the donor channel to be diffused across the membrane. Cy5-labeled exosome uptake by the cells was observed at both 12 and 24 h time points (Appendix A and Figure 4b).

### 2.5. Fluorescent EV-Producing Stable Cell Line Generation

To validate our system, we next analyzed the diffusion of EVs produced directly from cells through the Matrigel, and assessed their uptake by recipient cells. We first generated a stable cell line capable of producing EVs tagged with GFP for tracking live vesicle exchange on the chip. We used XPack lentivirus for this purpose, which encodes a GFP-fused peptide that targets the inner surface of EV membranes, including exosomes [33,34]. The lentivirus was transduced into U937 cells for production of a stable cell line (U937-XP). Stable U937-XP clones were generated by selection using puromycin treatment and identification of desirable clones based on GFP fluorescence intensity. Following expansion, the selected U937-XP clones were fluorescently imaged again and compared to wild-type U937 controls, validating strong expression of GFP fluorescence in the U937-XP clones in contrast to the background levels in control cells (Figure 5a).

The amount of fluorescence from vesicles secreted by U937-XPs was then quantified and compared to U937 vesicles. Conditioned exosome-free media (EFM) from both cell lines were used to collect 2000× *g* pellets (2K pellets), 10,000× *g* pellets (10K pellets), and sucrose-gradient-purified exosomes, which were then analyzed for relative GFP emission fluorescence intensity (Figure 5b). The 2K pellet and 10K pellet vesicles were also imaged (Appendix A). As expected, there was a significant fold increase in GFP fluorescence in each of the U937-XP vesicle populations (2K: 8.9x; 10K: 3.4x; exosomes: 3.2x), enabling fluorescence-based live observation of EV exchange [35].

### 2.6. Live EV Secretion, Diffusion, and Uptake

The U937-XP cells were then utilized for testing live EV secretion and diffusion across Matrigel, along with their subsequent uptake by recipient cells. Donor U937-XPs were injected into the donor channels of Matrigel-loaded chips (Figure 6a). The diffusion of fluorescent vesicles was confirmed by direct comparison of U937-XP donor cells and U937 cells (negative control), by imaging the matrix channel at the 24 h, 48 h, and 72 h time points (Figure 6b). By 48 h, the matrix became visibly fluorescent in the U937-XP donor chips. This coincides with the accumulation of fluorescent vesicles in the Matrigel (Appendix A). We then verified that cells in the recipient channel can internalize vesicles released by the donor channel cells. Based on the observed fluorescence accumulation timeline from Figure 6b, we prepared another Matrigel-loaded chip with U937-XP donor cells, but the recipient channel was left empty of any solution for 48 h in order to permit accumulation of the fluorescent EVs in the donor channel and the Matrigel, thus allowing higher EV concentrations for imaging of EV uptake. After 48 h, recipient U937s were introduced into the recipient channel, which also allowed entry of accumulated fluorescent EVs into this channel, and were imaged after a 4 h incubation period. The timeline of this experiment is illustrated in Appendix A. Successful uptake of fluorescent EVs by the recipient cells was observed in multiple cells (Figure 6c), confirming that secreted EVs can travel across the matrix channel and be subsequently internalized by recipient cells as part of a live EV exchange process. It is important to note that with the availability of an imaging system that allows long-term incubation of cells while performing constant high-frame-rate imaging, real-time EV diffusion and uptake can also be investigated in our chip system, where all channels are filled simultaneously.

## 3. Discussion

Our studies demonstrate the efficacy of our lab-on-a-chip system as a novel and effective assay for monitoring EV exchange. Our system works as intended—distinct cell populations can be injected into separate cell channels to accumulate and exchange EVs and small molecules, allowing for the observation of cell communication scenarios that more closely mimic the intricacy of a diverse in vivo cell environment. The working concentration of 8 mg/mL Matrigel injected into the chips allowed EVs of at least up to 250 nm to diffuse across. These included exosomes, which were observed to be internalized by recipient cells within the first 12 h. Therefore, our platform simulates in vivo EV exchange, during which cells send and receive exosomes and other EVs constitutively, in contrast to in vitro experimental procedures that involve treatment of an entire cell population with a certain amount of EVs at once. This can be an important consideration in some studies, where EV quantities that are used for in vitro analysis may not translate well to an in vivo setting in which physiological vesicle distribution and clearance influence effective EV availability and dosage [36]. Thus, the interconnected chip design with lengthy extended cell channels and live EV secretion can provide a closer portrayal of physiological tissue conditions and enable analysis of the functional consequences of EV and biomolecule communication between physically separated cells. In addition to providing a physiologically relevant platform for EV studies that bridges the in vitro and in vivo approaches, our system also provides some logistical advantages. For instance, the micron size of the cell channels of the chip drastically reduces the required volume of cells down to only 2–10 microliters. Moreover, the use of the chip obviates the need for costly and time-consuming purification of EVs, dramatically reducing the amounts of resources that are needed for various functional studies. Furthermore, our study using suspended U937 cells and the previous implementation of the chip that used adherent HUVEC cells [25] demonstrate the versatility of our system with respect to enabling the investigation of diverse cell types. Our results with PEGDA hydrogel also provide a strategy for preventing EV diffusion without affecting the passage of soluble factors, thus serving as a negative control for confirming that any observed effects are due to EV exchange. Further highlighting the versatility of our system, the ability to adjust PEGDA pore size should also provide a means of selective size exclusion of the EVs that can migrate across the Matrigel barrier, thus allowing for functional studies of EV subtypes that can be distinguished based on size differences.

Matrigel barriers and Transwell filter systems have been utilized previously, both to collect secreted EVs that have selectively diffused through the pores and accumulated in the supernatant [21], and to use pre-isolated EVs for cell migration studies [37]. Transwell systems have also been extensively used with or without supplementary Matrigel layers to investigate the role of certain exosome populations in cell migration [38,39,40,41,42]. Our microfluidic system circumvents the EV isolation process required for the aforementioned Transwell exosome functional strategies. In addition, it enables natural EV exchange between co-cultured donor and recipient cells, thus better recapitulating the physiological conditions. The Matrigel barrier in our chip also allows the observation of cell migration induced by exchange of bioactive molecules between cell populations in separate channels, as migratory cells can push through the Matrigel barrier. For instance, the accessory channel of the five-channel system can be used for conducting more elaborate experiments that also analyze the migratory behavior of cells towards the middle channel in response to the generation of stimulatory signals. Therefore, such studies can be performed without the need for prior isolation of biomolecules, similar to the utilization of the original chip design [25]. Other microfluidic technologies that have been developed in recent years to study cell communication between co-cultured populations rely on more complicated fabrication methods and the use of non-physiological barriers, such as filters or channels with reduced width, and also do not enable cell migration studies [43,44,45].

In future studies, we plan to optimize our capacity for large-scale production of the chips in order to allow their use by various researchers. The limited consistency of Matrigel loading, which requires precise manual pressure to be applied during chip preparation, prevents large-scale production of these chips from yet being feasible. However, due to the flexibility and ease of chip design modifications, we believe that such improvements are achievable. Additionally, we could manipulate this microfluidic chip system into a device for facile EV collection and purification. We also plan to demonstrate the functional assay capabilities of the chip by integrating it into our ongoing exosome research. For instance, the Hakami laboratory has shown that purified exosomes from cells infected with the Rift Valley fever virus (RVFV) can induce significant production of RIG-I-dependent interferon-beta (IFN-β) from naïve recipient cells, making them strongly refractory to infection with RVFV [4]. We can analyze this directly on the chip by injecting RVFV-infected cells into the top channel and naïve cells into the recipient channel. Furthermore, we can test whether exosomes that are, in turn, released by the recipient cells can also modulate the immune responses of naïve cells injected into the accessory (bottom) channel. Such studies will further confirm the utility of the chip and expand even further on the types and complexity of functional assays that can be performed.

## 4. Materials and Methods

### 4.1. Chip Design

A CAD drawing with multiple chip replicates was made using AutoCAD. The CAD drawing was submitted to MuWells (San Diego, CA, USA) to produce a chrome mask from the CAD drawing and fabricate a positive mold on a silicon wafer.

### 4.2. Chip Production/Optimization

Polydimethylsiloxane (PDMS; Sylgard 184) was mixed at a base-to-curing-agent ratio of 10:1. The silicon wafer was rinsed with isopropyl alcohol and dried with compressed air. The PDMS mixture was then poured onto the positive mold on the silicon wafer and degassed under vacuum for 30 min. Once all bubbles were removed, the PDMS was baked at 70 °C for 1 h. A large slab of PDMS was cut from the wafer region containing the templates. The PDMS slab was carefully removed and subsequently cut into individual chips. Holes were punctured into the inlets/outlets using 21- and 16-gauge blunt-end needles for the central/side and matrix channels, respectively. The PDMS chips and coverslips were then rinsed with ethanol and blown dry with compressed air to remove the PDMS debris. A handheld laboratory corona treater (BD-20AC, Electro-Technic Products, Chicago, IL, USA) was then used to plasma-activate the glass and PDMS surfaces for bonding, with settings at the maximum output voltage (~45 kV) for less than 5 s. The activated PDMS chips were placed on the activated coverslips, and air bubbles were removed. The chips were then baked at 70 °C for 1 h and transferred to a hot plate for incubation at 200 °C for 1 h in order to hydrophobize the PDMS. Subsequently, the corona treater was used to apply a plasma jet through the matrix channel in order to selectively activate it. To facilitate the selectivity of this activation, we inserted a copper ground wire at the other end of the matrix channel and blocked the other outlets with a slab of PDMS in order to use the pressure created by the plasma shock to funnel it through the matrix channel and prevent leakage into the middle and side channels. After activation, the chips were sterilized by exposure to UV light (365 nm) for 10 min. A volume of 3 μL of growth-factor-reduced Matrigel (Dow Corning, Midland, MI, USA) was then injected into the matrix channel and allowed to polymerize for 20 min at room temperature. After polymerization, the channels were coated with 0.05 mg/mL poly-D-lysine (Gibco, Waltham, MA, USA, REFA38904-01) for 1 h and subsequently rinsed and filled with PBS until ready for cell injection.

For use of poly(ethylene glycol) diacrylate (PEGDA) hydrogels, all preparation steps remained the same, except that the PEGDA solution was injected in place of Matrigel. PEG-400-DA was diluted in DI water to form a 20% *w*/*v* PEGDA solution. The photoinitiator, Irgacure (Sigma-Aldrich, St. Louis, MO, USA), was added to this solution at 0.5% *w*/*v* and vortexed for 30 s. This PEGDA/Irgacure solution was directly injected into the matrix channel. The injected matrix channel was polymerized under UV light for 15 min, and the chip was subsequently coated or filled with PBS.

### 4.3. Liposome Preparation

DMPC powder (850345P) and DOPE-PEG(2000)-N-Cy5 chloroform solution (880153C) were purchased from Avanti Polar Lipids (Alabaster, AL, USA); 70 nm and 250 nm liposome stock solutions (1 mg/mL) were produced by first dissolving DMPC in chloroform (Fisher Chemical, Waltham, MA, USA, C298-500) and then adding DOPE-PEG(2000)-N-Cy5 (99.9:0.1-DMPC:DOPE-PEG(2000)-N-Cy5) for future fluorescent visualization. The lipids were desiccated overnight in a vacuum chamber to form a lipid film. The next day, the lipids were dissolved in PBS (Gibco, 70011-044) and strongly vortexed. The lipid solution was then extruded through an Avanti Mini-Extruder (610000-1EA); 70 nm liposomes were passed through a 0.2 μm Nuclepore track-etched polycarbonate membrane filter (Whatman, Maidstone, UK, 800281), followed by a 0.05 nm filter (800308); 250 nm liposomes were passed through a 0.4 μm filter (800282) and then a 0.2 μm filter; 10 mm filter supports were used (Avanti Polar Lipids, 610014-1Ea). The size of the liposomes (Appendix A) was confirmed by dynamic light scattering (DLS) using a Zetasizer NanoSampler (Malvern, UK), and they were stored at 4 °C until further use.

### 4.4. Cell Line and Maintenance

U937 cells were purchased from the ATCC and maintained in RPMI 1640 medium supplemented with L-glutamine, 25 mM HEPES (Corning, 10-041-CV), and 10% exosome-free heat-inactivated fetal bovine serum (FBS) (Corning, Corning, NY, USA, 35-010-CV) that was prepared by ultracentrifugation at 100,000× *g* to remove FBS exosomes. This culture medium is designated as RPMI++ in the manuscript. Cells were incubated at 37 °C with 5% CO_2_ and split every 3–4 days. Cell count and viability measurements were conducted using a Luna Automated Cell Counter (Logos Biosystems, Annandale, VA, USA, L10001) in fluorescence measurement mode and using AO/PI dyes (VitaScientific, College Park, MD, USA, F23001). Cells within chips were also maintained in RPMI++, and the chips were kept inside a humidity container to alleviate long-term dehydration.

### 4.5. Puromycin Kill Curve

Stock puromycin was purchased from Sigma (P8833). A total of 3.5 × 10^4^ U937 cells in 200 μL of RPMI++ were added to 8 wells of a 48-well plate, and the following range of final puromycin concentrations was added to the wells: 0, 0.5, 1, 1.5, 2, 3, 5, and 10 μg/mL. The cells were counted again at 72 h before replenishing the media and puromycin. Following another 72 h of incubation, the cells were counted again. Based on the results, 1.5 μg/mL was selected as the optimal puromycin concentration (Appendix A).

### 4.6. U937-XP Cell Line Generation

As overviewed in Appendix A, 2.5 × 10^5^ U937 cells were spun down at 935× *g* for 5 min at room temperature. The cell pellets were washed with PBS and then resuspended in 200 μL of RPMI++ supplemented with 4 μg/mL polybrene (Specialty Media, Thermo Fisher Scientific, TR-1003-G) and transferred to a 48-well plate. XPack lentivirus (XPAK730VA-1), purchased from System Biosciences (Palo Alto, CA, USA), was added to the cells at MOIs of 0.3, 1, 5, and 10. A mock infection control well was also included. The cells were then spinoculated by centrifugation at 1340× *g* for 2 h at 32 °C. The supernatant was removed, and the cells were washed in serum-free RPMI with polybrene and centrifuged at 1455× *g* for 5 min. The supernatant was removed and the cell pellets were resuspended in fresh RPMI++ and transferred to a fresh well of a 48-well plate to incubate for 24 h at 37 °C and 5% CO_2_. After 24 h, the medium was replenished. At 72 h post-transduction, following confirmation of GFP fluorescence in the cells by imaging using an EVOS FL microscope (Invitrogen, Waltham, MA, USA), the culture medium was replaced with medium containing 1.5 μg/mL of puromycin. Every 72 h, the control cell population count was checked for total cell elimination, and the puromycin-containing RPMI++ was replenished. After 9 days, the control cells had completely died, and the remaining cell populations were checked for optimal GFP fluorescence using an EVOS FL microscope. The cells infected at an MOI of 10 had the most consistent and the brightest levels of GFP expression (Appendix A); therefore, 1.0 × 10^4^ cells from this cell population were transferred to a 96-well plate and were serially diluted 2-fold vertically and then horizontally down the entire columns and rows. Several wells were chosen for having the smallest numbers of clones and then grown for 4 days before being transferred to a 6-well plate and allowed to grow for another 48 h. They were selected again based on qualitative GFP expression. The selected clones were then transferred to a new plate and grown to a higher density to be used in the preparation of frozen stocks for later experiments.

### 4.7. Extracellular Vesicle (EV) Purification

In addition to filter sterilization of the final purified sample, all solutions and tubes used during this purification procedure were also sterilized as an extra precaution. U937 cells were seeded at 5 × 10^5^ cells/mL in 100 mL of exosome-free medium (EFM). At 72 h, the cells were spun down at 600× *g* for 5 min. The supernatant was collected and spun down at 2000× *g* for 20 min, and the resulting supernatant was centrifuged again at 10,000× *g* for 30 min. The 2000× *g* and 10,000× *g* pellets (2K and 10K pellets) were resuspended in 100 μL of PBS for further analysis, and the remaining supernatant from the 10K spin was filter-sterilized through a 0.2 μm bottle-top filter (VWR, 10040-468) and subsequently spun down at 167,000× *g* for 2 h to pellet the exosomes. The supernatant was removed, and the exosome pellet was resuspended in PBS and centrifuged again at 167,000× *g* for 2 h. The exosomes were then resuspended in 400 μL of PBS and were either left unlabeled or were fluorescently labeled by incubating them with 25 μL of DOPE-PEG(2000)-N-Cy5 at 37 °C. Both unlabeled and Cy5-labeled exosomes were then purified further side by side, by overlaying them on top of a stepwise sucrose (VWR, 0335-1KG) density gradient containing density fractions of 1.25, 1.167, 1.149, 1.103, 1.064, and 1.031 g/mL followed by ultracentrifugation at 167,000× *g*. Through numerous (>30) biological repeats and characterizations of this purification procedure, we have previously verified that the exosomes recovered from U937 cells migrate to the 1.149 and 1.103 g/mL density fractions. Therefore, these two fractions were combined, diluted by the addition of 20 mL of PBS, and centrifuged at 167,000× *g* to remove the sucrose. The pellet was then resuspended in 100 μL of PBS and filter-sterilized using a 0.20 μm sterile syringe filter (VWR, 28145-477). Following addition of the Halt Protease Inhibitor Cocktail (Thermo Scientific, Waltham, MA, USA, 1862209) at 1× final concentration, the exosomes were stored at −80 °C until further use. The size and quantity of U937-XP exosomes were determined (Appendix A) using ZetaView nanoparticle tracking analysis (Particle Metrix GmbH, Inning am Ammersee, Germany), as previously described [4].

### 4.8. EV Fluorescence Measurement and Analysis of the EV Pellets

Fluorescence readings were taken for the 2K pellets, 10K pellets, and purified exosomes from both U937 and U937-XP cell lines using a Tecan Safire 2 multi-detection microplate reader and analyzed using Magellan software (Tecan, Männedorf, Switzerland). A volume of 30 μL of each vesicle population was used for analysis; the protein concentrations of the 2K pellets, the 10K pellets, and the purified exosomes were on average 400 μg/mL, 30 μg/mL, and around 250 μg/mL, respectively. The experiment was performed in 3 biological replicates. Unpaired Student’s *t*-tests were performed for statistical analysis and comparison between U937 and U937-XP vesicles.

### 4.9. Vesicle and Cell Injections into Chips

A 5 μl volume of either purified exosomes, liposomes (1 mg/mL), or 500 nm Fluoresbrite YG carboxylate microspheres (Polysciences, Warrington, PA, USA, 17152-10) (25 μg/mL) was injected into the chips by pipetting into the inlet ports of the donor channels. A 5 μL volume from a suspension of 2 × 10^7^ cells/mL was injected into the recipient cell channels at a flow rate of 0.5 μL/min using a Fusion 200 (Chemyx, Stafford, TX, USA) syringe pump, a 250 μL syringe (Hamilton, Reno, NV, USA 81175), and 24-gauge PTFE tubing (Component Supply, Sparta, TN, USA, TWTT-24-C) tightly inserted into the injection port. For cell injection into donor channels, a 10 μL volume of cell suspension at 2 × 10^7^ cells/mL was injected at a flow rate of 1 μL/min. Excess solution was wiped away from the outlet port.

### 4.10. Fluorescent Imaging

The cells imaged on slides were first fixed by incubation in 2% PFA (Macron Fine Chemicals, Avantor, Center Valley, PA, USA, H121-08) in PBS for 20 min at room temperature, followed by two PBS washes. The cells were then mounted onto glass slides using DAPI–Fluoromount G (SouthernBiotech, Birmingham, AL, USA, 0100-20). Fluorescent imaging of the fixed cells and chips was performed using a Ti2 series inverted microscope (Nikon, Tokyo, Japan) outfitted with a 100X oil immersion objective, using standard bright-field and Cy5 (λEX: 625, λEM: 670 nm), DAPI (λEX: 360 nm, λEM: 460 nm), or GFP (λEX: 480 nm, λEM: 535 nm) filters. Z-stack imaging, merged channels, 3D deconvolution, and stitching were performed using Nikon’s NIS-Elements AR software.

## 5. Conclusions

In this study, we developed a novel and versatile microfluidic platform to enable future mechanistic investigations of real-time intercellular communication—including EV exchange studies—across various disciplines. In addition to Matrigel, our platform can also accommodate other diffusion barrier substances to allow for tailor-made studies, including hydrogels of variable pore size and composition. The system can also be scaled to accommodate various study needs, as we demonstrated by adapting the system from a three- to a two-cell lane design. The ability to selectively inject the diffusion barrier (e.g., Matrigel or PEGDA hydrogel) within a single channel is a key feature to enable the facile preparation of these chips, which should facilitate their adoption and use. Further development and study-specific optimization of the diffusion barrier and rib length will also enable control over the timescale of vesicle communication. In addition, the use of PEGDA or other hydrogel barriers with various pore sizes should provide a means of selecting EV subtypes that migrate across, or allow their retention, enabling functional studies of EV subtypes that can be distinguished based on size differences.

## Figures and Tables

**Figure 1 ijms-23-03534-f001:**
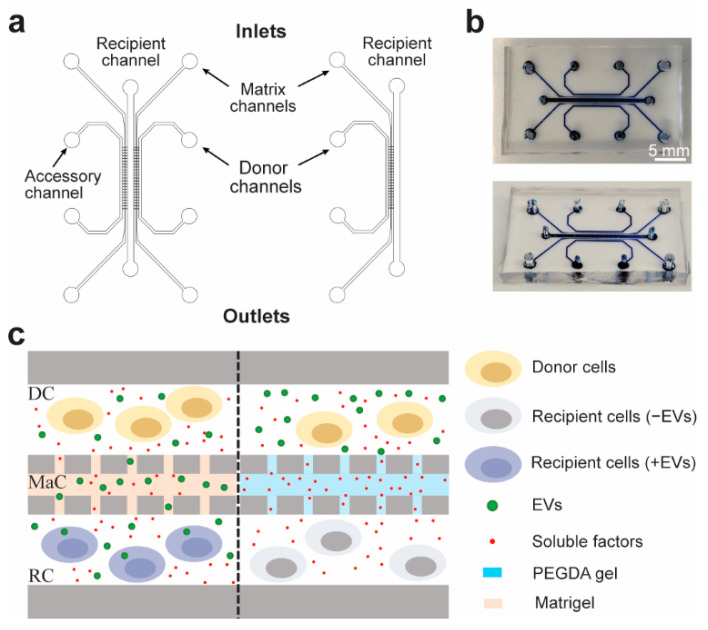
Microfluidic chip design and conceptual use: (**a**) Schematic of the 5-channel (**left**) and 3-channel (**right**) microfluidic chip designs depicting the matrix, donor, recipient, and accessory channels. (**b**) Fully assembled chip loaded with blue ink to visualize the channels. (**c**) Schematic of the donor and recipient cells in the donor channel (DC) and recipient channel (RC). The matrix channel (MaC) can be filled with either Matrigel or PEGDA to permit size-based diffusion of either EVs and soluble factors or only soluble factors, respectively.

**Figure 2 ijms-23-03534-f002:**
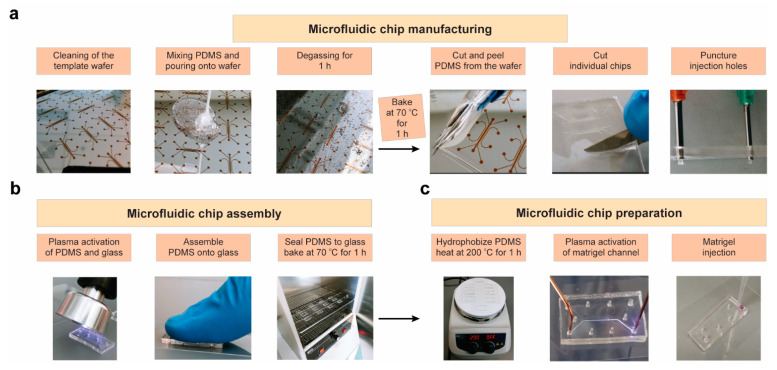
Full workflow for microfluidic chip fabrication, assembly, and preparation. Steps for (**a**) manufacturing the chips, (**b**) chip assembly, and (**c**) activating the matrix channel for injection are shown.

**Figure 3 ijms-23-03534-f003:**
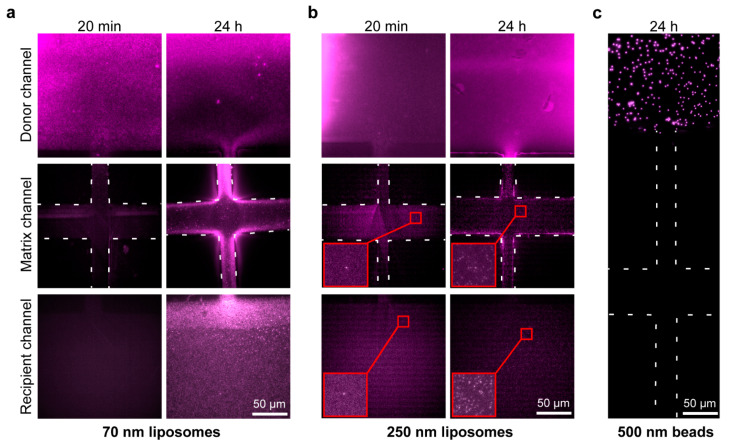
Characterization of particle diffusion: The size selectivity of Matrigel at 8 mg/mL was tested by diffusion of different-sized particles. Matrigel-loaded chips were injected in their donor channels with (**a**) 70 nm or (**b**) 250 nm DOPE-PEG(2000)-N-Cy5-labelled liposomes, and (**c**) 500 nm Fluoresbrite^®^ (Polysciences, Warrington, PA, USA) yellow–green fluorescing polystyrene beads, presented here as a false-color image for consistency with panels (**a**,**b**). The chips were visualized at 20 min and 24 h after injection, using (**a**,**b**) Cy5 or (**c**) GFP emission filters to confirm particle diffusion from the donor channel across the matrix channel and into the recipient channel (matrix channel ribs are outlined with white dashed lines. Insets are outlined in red color, and have a width of 10.80 μm. Original magnification = 1000×).

**Figure 4 ijms-23-03534-f004:**
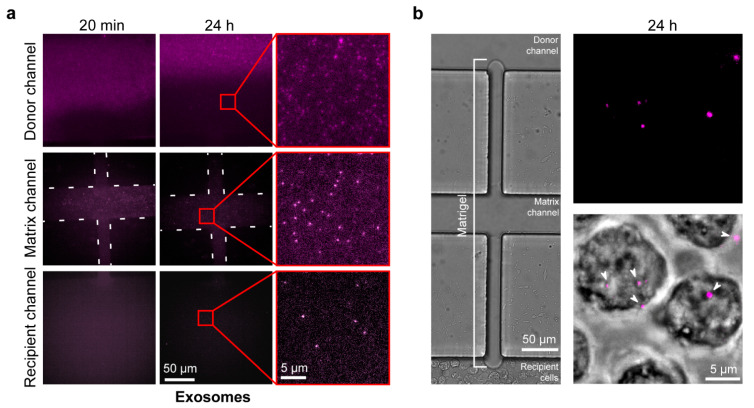
Verification of exosome diffusion across the Matrigel and uptake by the recipient cells in the central channel: The ability to observe and investigate exosomes using our Matrigel chip assays was confirmed by demonstrating diffusion of labelled exosomes across the Matrigel into recipient cells. (**a**) A Matrigel-loaded chip was injected with DOPE-PEG(2000)-N-Cy5-labelled U937 exosomes into the donor channel. The donor channel, matrix channel, and recipient channel were visualized at 20 min and 24 h after injection, using a Cy5 emission filter to observe exosome diffusion (Matrix channel ribs are outlined with white dashed lines. Insets are outlined in red. The insets were contrast-enhanced for visual clarity. Original magnification = 1000×). (**b**) A Matrigel-loaded chip was injected with DOPE-PEG(2000)-N-Cy5-labelled U937 exosomes in the donor channel and recipient U937 cells in the recipient channel. Chips were visualized at 24 h after injection using bright-field and Cy5 emission filters showing (**Left**) no interchannel cell migration through the Matrigel and (**Right**) uptake of exosomes by recipient cells (arrowheads point at imaged exosomes that are either inside the cells or are interacting with the cells on the surface. Original magnification = 1000×).

**Figure 5 ijms-23-03534-f005:**
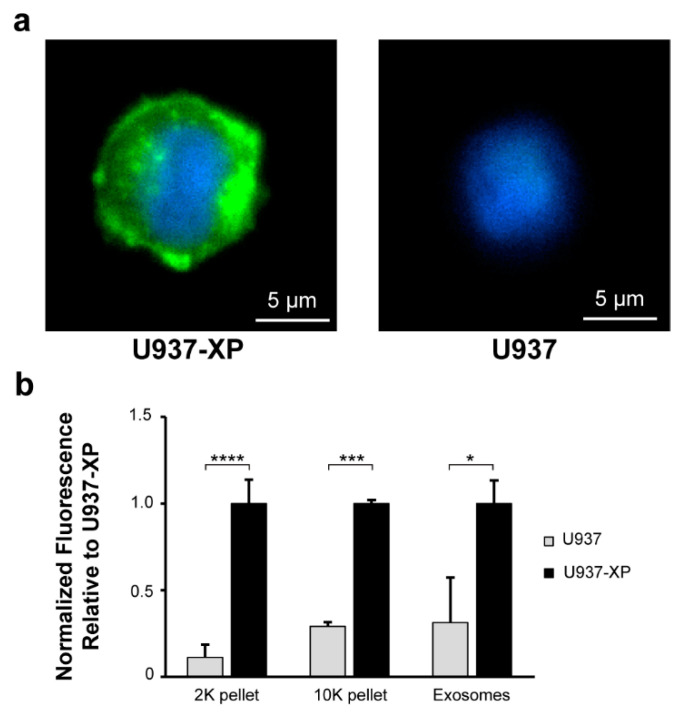
Generation of a stable U937 cell line for the production of fluorescently tagged EVs: To study live EV exchange, a U937 cell line was transduced with XPack lentivirus to generate a new stable cell line (U937-XP) for the production of fluorescent EVs. (**a**) U937-XP and U937 cells were visualized using GFP and DAPI emission filters following fixation and mounting with DAPI-containing mounting medium (original magnification = 1000×). (**b**) Relative fluorescence values for vesicle populations recovered from U937 and U937-XP cells are shown. Fluorescence values show the means of 3 biological repeats for emission levels at 507 nm after excitation at 455 nm, adjusted for PBS blank and normalized relative to the mean value for U937-XP measurements (Student’s *t*-test; * *p* ≤ 0.05, *** *p* ≤ 0.001, **** *p* ≤ 0.0001; error bars represent standard deviation (S.D.)).

**Figure 6 ijms-23-03534-f006:**
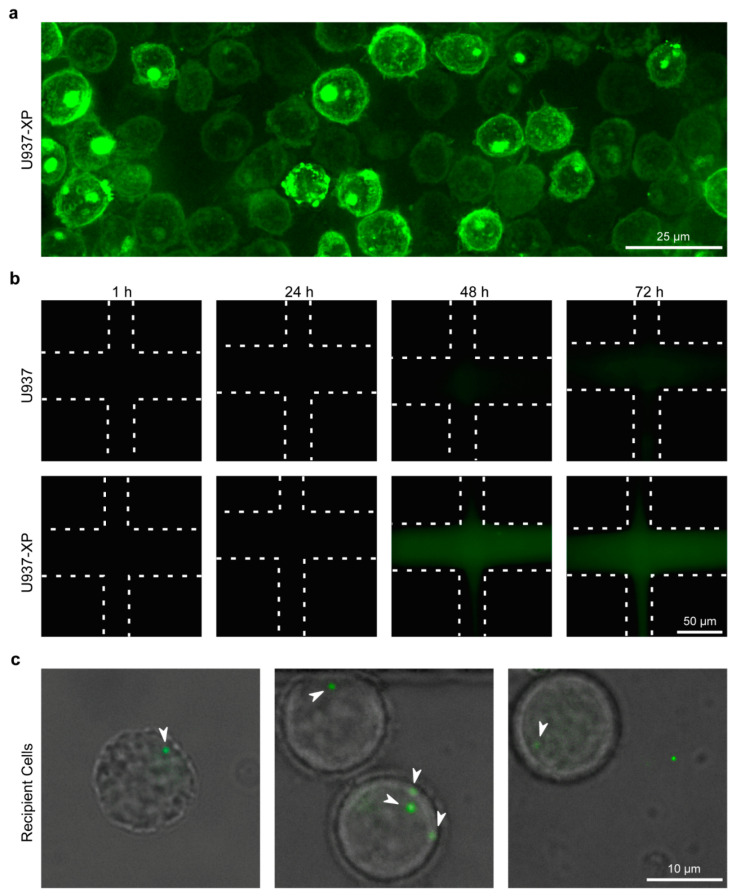
Live EV exchange on the microfluidic chip: To verify the ability of our chips to serve as reliable platforms for live EV exchange, we monitored transmigration of secreted EVs across the Matrigel channel and their internalization by cells in the recipient channel. (**a**) A Matrigel-loaded chip was injected in the donor channel with U937-XP cells, and the cells were visualized 1 h later using a GFP emission filter (original magnification = 1000×; 3D deconvoluted.). (**b**) Matrigel-loaded chips were injected into the donor channel with either U937-XP cells or U937 cells as a negative control, and the Matrigel-loaded matrix channel was visualized at 1 h, 24 h, 48 h, and 72 h for the presence of fluorescence from diffusing vesicles using a GFP emission filter (Matrigel channel ribs are outlined with dashed lines. Original magnification = 1000×). (**c**) A Matrigel-loaded chip with U937-XP cells injected into the donor channel, and with an empty recipient channel, was incubated for 48 h. Subsequently, U937 cells were injected into the recipient channel and, 4 h after the cell injection, the internalization of U937-XP-secreted EVs by recipient cells was visualized using bright-field and GFP emission filters (arrowheads point at imaged EVs inside the recipient cells. Magnification = 1000×).

## Data Availability

All of the data generated in this study will be made available upon reasonable request to the corresponding authors.

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
