# Peer review of "A Microfluidic Platform to Monitor Real-Time Effects of Extracellular Vesicle Exchange between Co-Cultured Cells across Selectively Permeable Barriers"

_ijms, 2022, doi:10.3390/ijms23073534_

Round 1
Reviewer 1 Report
In the manuscript “A Microfluidic Platform to Monitor Real-time Effects of Extracellular Vesicle Exchange between Co-cultured Cells Across Selectively Permeable Barriers”, Mason and Bush et al. describe and test a chip developed by them to monitor Extracellular Vesicle (EV) exchange between two cell populations separated by a semisolid matrix, which pore size can be customized. I agree that that this system is practical as shown by the authors and offers some advantages over the classical studies involving EVs like exosomes.
Nevertheless, the manuscript can be improved by revising some aspects indicated below.
1- Figures should not be referenced in the introduction and methods sections.
2- Regarding the chip design, I did not understand whether the exosomes will migrate by diffusion only (horizontal chip with the channels at the same level) or whether gravity will help (vertical chip with one channel on top – donor channel – and another one at the bottom – recipient channel, Figure 1A).
3- Why do the authors show a 5-channel chip in Figure 1 (main) but use a 3-channel chip (Figure S1) in their experiments?
4- In lines 156-159 it is written “Both vesicle populations were able to diffuse through within the 24-hour period. As expected due to their smaller size, the 70 nm vesicles diffused through more rapidly than the 250 nm when compared at the 24-hour time points (Figure 3a and 3b, Recipient channel)”. Only by analyzing the figure, I cannot say that at 20 minutes, there are more 70 nm vesicles in the recipient channel when compared to 250 nm vesicles in the respective recipient channel.
5- In lines 249 to 252 it is written “based on the observed fluorescence accumulation timeline from Figure 6b, we prepared another Matrigel-loaded chip with U937-XP donor cells, but the recipient channel was left empty of any solution for 48 hours to permit accumulation of the fluorescent EVs in the donor channel and Matrigel, thus allowing higher EV concentrations for imaging of EV uptake”. Did the authors try to do it by filling simultaneously all the channels? Should the chip be used in this sequential way when testing live EV secretion and diffusion across Matrigel, and their subsequent uptake by recipient cells?
6- The authors should discuss also whether the chip allows to culture adherent cells or only cells growing in suspension.
7- In lines 394 to 395 it is written “cells within chips were also maintained in RPMI++, and the chips were kept inside a humidity container to alleviate long term dehydration”. RPMI++ appears several times in the Methods section; what does it mean?
8- The title of a table should be on top (Table S1).
Reviewer 2 Report
Mason et al. described a lab-on-chip system to filter the extracellular vesicles (EVs) from larger EVs, e.g., apoptotic bodies and bacteria. The system can potentially be used to purify exosomes and also to study exosome-cell interactions. There are several queries that I hope the authors could look into and address:
- Figures 3 and 4: Can the fluorescence signal emitted by the liposomes and exosomes, respectively, in the recipient channel be quantified? It would be great if the signal intensity in the matrix channel can also be quantified.
- Figure 6: The result appears contradictory. The fluorescence is hardly seen in the matrix channel at 24h but the authors could detect the uptake of exosomes after 4h. Perhaps the signal intensity was just low at 24h and was not visible on the digital print. The authors should consider studying the uptake of the EVs beyond 4 hours to find out whether the amount of the EVs passing through the matrix commensurates with the fluorescence intensity in the matrix channel and in the recipient cells over time.
- Can the authors comment in the Discussion whether their lab-on-chip system can be used to study adherent cells, at least in the recipient channel?
- Perhaps, the Conclusions can be moved to before the Materials and Methods section.
- Line 332: tun. Do the authors mean "turn"?
- Line 352: How long was the PDMS treated for? What was the power setting of the corona treatment?
- Line 418: to for. Please correct. It does not sound right.
Round 2
Reviewer 2 Report
The authors have addressed all my queries. I am satisfied with it.